# Fast and Robust Asynchronous Rendezvous Scheme for Cognitive Radio Networks

## Yongchul Kim

Department of Electrical Engineering, Korea Military Academy, 574 Hwarangro, Nowon-Gu, Seoul 01805, Korea; kyc6454@kma.ac.kr; Tel.: +82-10-7439-0903

**Abstract:** The rendezvous process is considered a key operation that allows a secondary user (SU) to access an unused authorized spectrum in cognitive radio networks (CRNs). Most existing works focused on fast guaranteed rendezvous without considering a sophisticated jamming attack environment. In this paper, I propose a fast and robust asynchronous rendezvous scheme that can improve robustness against jamming attacks under symmetric asynchronous environments in which all SUs have the same available channels. Unfortunately, in CRNs, each SU can have a different number of available channels due to their relative position to primary nodes (PUs). Therefore, I extend my fast and robust asynchronous rendezvous scheme (FRARS) to a general asymmetric scenario while preserving robustness against jamming attacks. I derive the maximum rendezvous time (MTTR) of my new algorithm and the upper bound of the expected TTR (ETTR) and compare it with the state-of-the-art algorithms such as jump-stay (JS) and Enhanced jump-stay (EJS). My numerical results show that the performance of the proposed technique is better than that of JS and EJS in terms of MTTR and ETTR. Also, the performance will be more significant when there are security concerns about a sophisticated jamming attack.

**Keywords:** cognitive radio networks (CRNs); jamming attack; blind rendezvous; channel-hopping (CH)

---

## 1. Introduction

Unlicensed spectrum bands that can be used without complying with the regulations applicable to licensed services have already become globally overcrowded. In contrast, large portions of the licensed spectrum bands are severely underused as shown in federal communications commission report [1]. To overcome this problem, a method has been widely used in which a secondary user (SU) can use an authorized spectrum that is not in use opportunistically without interfering with primary users (PUs). Cognitive radio networks (CRNs) have been recognized as a promising paradigm to improve the spectrum efficiency of wireless communications [2–4]. Devroye et al. [3] highlight some of the recent information theoretic limits, models, and design of these promising CRNs. Liang et al. [2] provide a systematic overview of CRNs and communications by looking at the key functions of the physical, medium access control, and network layers involved in a CR design and how these layers are crossly related. In CRNs, a pair of SUs wishing to communicate with each other must establish a link, i.e., two SUs must simultaneously exchange handshake information on a common channel and this process is called the rendezvous process. Most existing works [5,6] focus on implementing a rendezvous process by using dedicated common control channel (CCC) for the sake of simplicity. However, CCC is probably occupied by a PU that is an incumbent license holder of a frequency band, and maintaining a CCC may result in a bottleneck problem as well as creating a single point of failure. Therefore, I study a different scheme to rendezvous without using a centralized controller. In this approach, each SU uses a channel-hopping (CH) sequence with guaranteed rendezvous during the CH sequence. This process is known as blind rendezvous. In most CH algorithms, time is divided

into slots of constant size, and the number of time slots required for two SUs to achieve a rendezvous after all SUs start the rendezvous process is defined as rendezvous time (TTR). If a pair of SUs achieve successful rendezvous on the same channel at the same time, they can exchange control messages and select one of the commonly available channels to start data transmission. The MTTR and the ETTR are two important metrics commonly used to evaluate performance of proposed CH schemes.

In CH algorithms, each SU in the CRN periodically senses the spectrum of the PU to determine the available channels for the rendezvous process. Each SU can have a different set of channels because geographic locations of SUs are different. If all SUs have the same available channels, this is called a symmetric system, otherwise asymmetric system, i.e., different SUs have different numbers of available channels. There has been much work in the literature to guarantee blind rendezvous, but most work focuses on symmetric scenarios due to time synchronization constraints. Moreover, few of those works consider security attacks such as jamming. One of the most notable works, as addressed in a recent survey paper [7], is the jump-stay (JS) rendezvous algorithm [8]. The same author revises the original configuration and suggests an improved jump-stay (EJS) algorithm [9] to improve performance for the asymmetric scenario. It is known that the EJS is one of the best blind rendezvous algorithms for CRNs based on non-deterministic CH sequences and the guaranteed rendezvous times for both symmetric and asymmetric models. The basic idea in both JS and EJS algorithms is to generate CH sequences based on jump and stay patterns to ensure blind rendezvous without time synchronization. These schemes can be applied to the rendezvous of multi-user and multi-hop scenarios. Nevertheless, those are significantly vulnerable to a hostile environment, i.e., the CH sequences can be easily detected and jammed by the channel-detecting jamming attack (CDJA) [10]. This paper analyzes the limitations of JS and EJS by revisiting the CDJA and propose a fast and robust asynchronous rendezvous scheme (FRARS) to overcome those limitations. I also include a theoretical analysis as well as extensive simulations under CDJA to demonstrate that the proposed technique is superior to other recently proposed methods, including JS and EJS.

The remainder of this paper is organized as follows. The following section reviews related tasks in the CRN. In Section 3, I present well-known CH schemes such as JS and EJS as well as channel-detecting jamming attack model. Section 4 presents my proposed FRARS for both symmetric and asymmetric systems. The numerical results and evaluations are described in Section 5. Section 6 is the conclusion of the paper.

## 2. Related Work

Liu et al. [7] provides a comprehensive review of CRN existing rendezvous algorithms and categorizes those algorithms into two groups—centralized and decentralized. In a centralized system [11,12], a centralized controller manages the rendezvous of all SUs in the networks. Guerra et al. [13] introduced a systematic construction of common channel-hopping rendezvous strategy to guarantee that every node should be able to rendezvous in all common available channels. However, these algorithms have limitations discussed in the previous section. In a decentralized system, each SU attempts rendezvous without any help from a controller. Recently, many studies have focused on decentralized rendezvous system without using common control channel, i.e., blind rendezvous. A typical method of blind rendezvous is the CH technique. The taxonomy of rendezvous algorithms [7,14] shows three categories for blind rendezvous; random algorithms, synchronous algorithms, and asynchronous algorithms. The asynchronous algorithms can be further classified into two sub-categories—a symmetric model and an asymmetric model. A purely random [15] or an improved random algorithm [16] provide a trivial CH algorithm where SUs rendezvous by chance. However, these random algorithms cannot guarantee a bounded TTR. Both synchronous and asynchronous algorithms provide guaranteed rendezvous. Several synchronous algorithms are proposed with an assumption of global time synchronization [17–19]. Bahl et al. [17] introduced a Slotted Seeded Channel Hopping to increase the capacity of an IEEE 802.11 network and Krishnamurthy et al. [18] proposed a deterministic approach in which each SU can dynamically

calculate a common set of channels globally. Bian et al. [19] introduced two quorum-based CH methods (M-QCH and L-QCH) that can guarantee rendezvous between time-synchronized SUs. These synchronous systems are usually unfeasible for blind rendezvous and the impact of a jamming attack can be significant.

Most of the recently proposed algorithms [20–25] consider asynchronous scenarios without time synchronization. DaSilva and Guerreiro [20] proposed a generated orthogonal sequence (GOS) algorithm that uses interspersed permutation channels to guarantee rendezvous. Theis et al. [21] introduced modular clock and modified modular clock algorithms in which each SU generates its CH sequence from pre-defined modulo operations. Liu et al. [22] proposed a ring walk (RW) algorithm by using the idea of velocity. Thus, the rendezvous is achieved when the SU with lower velocity is caught by the SU with higher velocity. Yang et al. proposed two significant algorithms, namely deterministic rendezvous sequence (DRSEQ) [23] and channel rendezvous sequence (CRSEQ) [24], which provide fast asynchronous rendezvous under symmetric and asymmetric models, respectively. The performance of CRSEQ is good under the asymmetric model but it does not perform well under the symmetric model. Pu et al. [25] designed an efficient dynamic rendezvous algorithm when the status of the channels varies dynamically for both synchronous and asynchronous users. Among the rendezvous algorithms that are applicable to asymmetric scenario, the JS [8] and EJS [9] are known to have the overall best performance. However, most of the aforementioned algorithms including the JS and EJS are significantly vulnerable to a sophisticated jamming attack. In this paper, I propose a FRARS algorithm to overcome the vulnerability problem by employing a randomized permutation technique while preserving fast guaranteed rendezvous. I also derive the upper bounds on MTTR and ETTR of the FRARS for both symmetric and asymmetric systems.

## 3. Preliminary

This section presents two well-known modular-based asymmetric rendezvous schemes, JS and EJS algorithms. I then introduce a CDJA model that demonstrates how effectively the JS and EJS algorithms are attacked. I use all the terminologies defined in Lin's work [8].

### 3.1. Jump-Stay Algorithm

Lin et al. [8] proposed the JS algorithm that provides a guaranteed rendezvous for both symmetric and asymmetric models in the CR networks. The considered CRN consists of $K$ SUs ($K \geq 2$), who coexist with several PUs. The licensed spectrum can be divided into $M$ non-overlapping channels $C = \{c_1, c_2, ..., c_M\}$, where $c_i$ denotes the $i$th channel. Let $C_k \subseteq C$ denote the set of channels available to User $k$ ($k = 1, 2, ..., K$) and let $G$ be the number of common available channels of the users, i.e., $G = |\bigcap_{i=1}^{K} C_i|$. In the symmetric model, all SUs have the same available channels, i.e., $C_i = C_j (1 \leq i, j \leq K)$. For simplicity, I assume $C_i = C_j = C$. Each SU generates its CH sequences in rounds and each round consists of a jump pattern and a stay pattern. To generate the sequences in a round, each SU selects three parameters: $P$ which is the smallest prime number greater than $M$, an index $i_0$ from $[1, M]$ for the starting channel index and a non-zero number $r_0$ from $[1, M]$ for the step length parameter. Using this information, each round takes $3P$ time slots that consists of two jump patterns for $2P$ time slots and one stay pattern for $P$ time slots. In the jump pattern, the SU starts with index $i_0$ and keeps jumping in $[1, P]$ with step length $r_0$ by using the modulo operations on $P$. In the following stay pattern, the SU just stays on channel $r_0$ which is the step length of this round. Every round (i.e., every $3P$ time slots), step length $r_0$ is changed to the next number in $[1, M]$ in round-robin fashion. The starting index $i_0$ will remain the same in all rounds for the symmetric case but for the asymmetric scenario it will be changed to the next number in every $6MP$ time slots. The JS algorithm guarantees the rendezvous and performs well under both the symmetric and asymmetric models. The MTTR values of JS for symmetric and asymmetric models are $3P$ and $6MP(P - G)$, respectively. Figure 1 shows a rendezvous example of the JS scheme, where $M = 4$ and $P = 5$. User 1 starts its sequence with $i_1 = 2$ at $t = 1$ and jumps to

the next channel with the step length $r_1 = 1$. The other user can start at any time within $3P$ time slots because there is no time synchronization, thus User 2 starts its sequence with $i_2 = 3$ at $t = 5$ and jumps to the next channel with the step length $r_2 = 2$. Therefore, at time slot $t = 8$ the two users jump to the same channel, Channel 4.

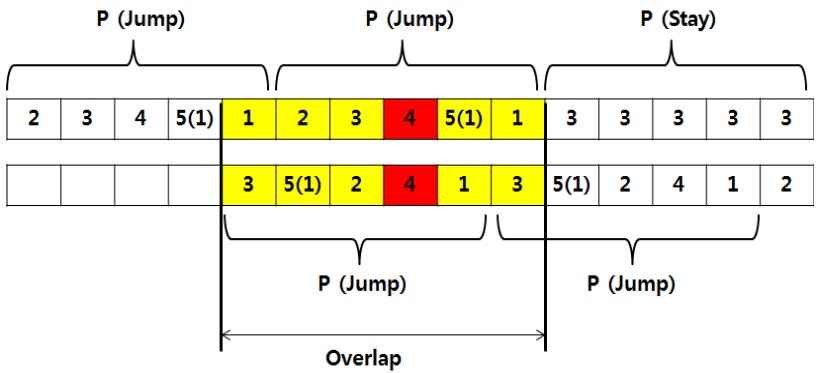

**Figure 1.** An example of symmetric JS Algorithm.

## 3.2. Enhanced Jump-Stay Algorithm

The same authors improve the performance of JS by proposing the EJS algorithm that can reduce the upper bounds of both MTTR and ETTR values from $O(P^3)$ to $O(P^2)$ under the asymmetric model, while keeping the same order of rendezvous times of the symmetric case. Unlike JS, EJS's jump pattern lasts $3P$ time slots, i.e., each round takes $4P$ time slots which consists of three jump patterns for $3P$ time slots and one stay pattern for $P$ time slots. In addition, the starting index $i$ is randomly selected from $[1, P]$ and switched to the next number every round ($4P$ time slots) in the round-robin fashion, while the step length $r$ remains the same in every round after randomly selecting from the available channels for both symmetric and asymmetric models. Therefore, the channel $c$ of the jump pattern is determined as follows. $c = ((i + t \times r - 1) \mod P) + 1$ where $c$ is the channel of time $t = t \mod 4P$. If $c$ is greater than $M$, then it is remapped to $[1, M]$ (i.e., $c = ((c - 1) \mod M) + 1$ for $c > M$). For the asymmetric model in EJS, additional replacement process is performed to increase the probability of rendezvous. That is to say, if $c_j$ is the computed channel in the sequence of User $k$ and it does not belong to $C_k$, then it is replaced by $((j - 1) \mod |C_k|) + 1)th$ channel in $C_k$. For example, given $C_k = \{c_1, c_3, c_4\}$ and $j = 5$, since $c_5$ does not belong to $C_k$, it is replaced by $c_3$ (Here, $((j - 1) \mod |C_k|) + 1) = 2$ and $c_3$ is the $2nd$ channel in $C_k$). Figure 2 illustrates an example rendezvous of the symmetric EJS system where $M = 4$ and $P = 5$. The rendezvous process is very similar to the JS system, except the length of one round of EJS is longer than that of the JS system. For the asymmetric scenario, two users' CH sequences under EJS scheme are shown in Figure 3 when $M = 2$ and $P = 3$ without substitution process. User 1 as a sender has two available channels $\{c_1, c_2\}$ and User 2 as a receiver has only one available channel $\{c_2\}$. Two different step lengths for User 1 are considered in this example. When the step lengths of the sender are 1 and 2, the sender and receiver rendezvous at slot numbers 14 and 10, respectively. The upper bounds of MTTR for symmetric and asymmetric models are proved to be $4P$ and $4P(P + 1 - G)$ time slots, respectively in Lin's work [9].

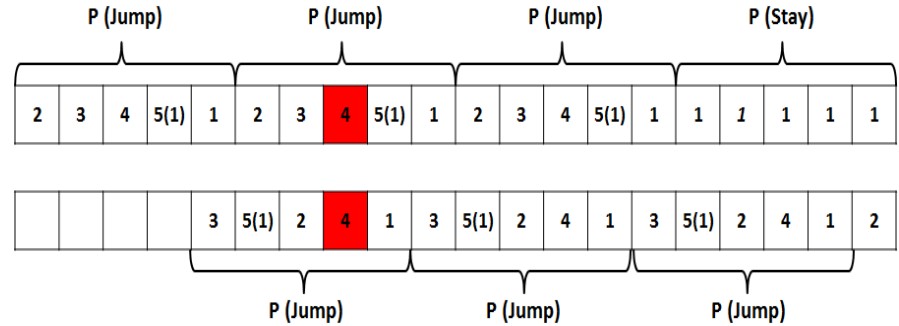

**Figure 2.** An example of symmetric EJS Algorithm.

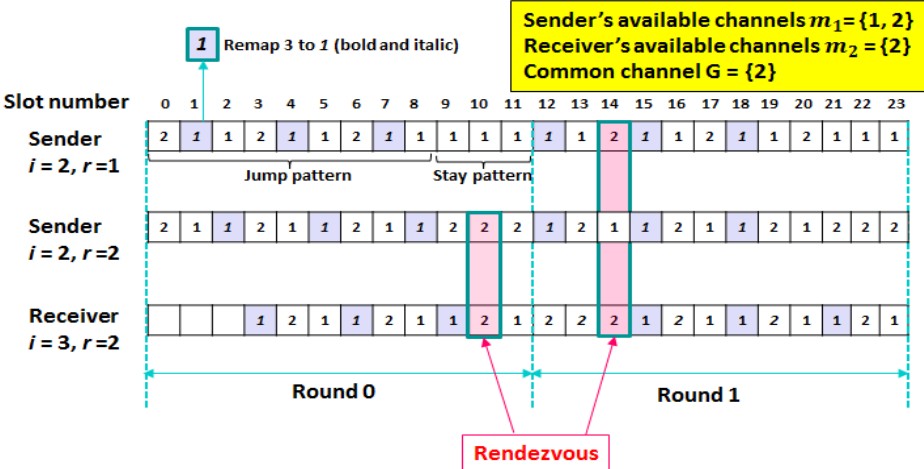

**Figure 3.** An example of asymmetric EJS Algorithm.

### 3.3. Channel-Detecting Jamming Attack

To show the vulnerability of JS and EJS against a sophisticated jamming attack, I present a CDJA model in this section. This jammer can dramatically decrease the probability of rendezvous for both JS and EJS schemes. Since the EJS is an enhanced algorithm of the JS, I focus on how to determine the CH sequence of a sender for EJS's symmetric and asymmetric scenarios. Channel-detection jammers have similar functions to a legitimate user, but assumes that it can listen on two channels at the same time by using two transceivers. Although one listening-channel jammer can detect the CH sequences, it is much more difficult and the cost of adding one more listening channel is very low. I also assume a normal situation where a jammer is already waiting on the network before communication between the two SUs begins. When there is no jamming attack, the symmetric EJS scheme guarantees that two SUs can rendezvous in up to $4P$ time slots as described in the previous subsection. However, when the jammer selects two channels and listens until the sender's activity is detected, the jammer can find two channels during $P$ time slots and the step length must be able to be calculated. Therefore, the jammer can find out all the remaining CH sequences after an average $\lfloor \frac{(P+1)}{2} \rfloor$ time slots. In other words, two SUs must rendezvous within an average $\lfloor \frac{(P+1)}{2} \rfloor$ time slots in the CDJA attack situation. Thus, the rendezvous probability will significantly decrease. For the asymmetric EJS system, the channel-detecting jammer uses the replacement algorithm as well as modular-based properties. Also, I assume that jammers can find the available channels of the sender by using spectrum-sensing techniques. When the jammer is waiting for the sender's signals, it selects two random listening channels among the sender's available channels. The jammer can then find two channels within the first $P$ time slots and be able to compute the step length in a similar way to the symmetric scenario. However, the calculated step length may be incorrect because the detected channels can be either non-replaced or replaced channels. To verify the step length, the jammer needs to listen on more channels to make sure the step length is correct.

Since the initial channel $i$ increases for consecutive rounds and the step length $r$ is fixed in EJS, all subsequent CH sequences can be easily calculated by the jammer. The detailed procedures of detecting CH sequences for both symmetric and asymmetric scenarios are presented in Oh's work [10].

## 4. Proposed Channel-Hopping Scheme

I propose a FRARS algorithm with robust performance against jamming attacks in symmetric and asymmetric scenarios and derive the MTTR and ETTR of the proposed FRARS.

### 4.1. FRARS for Symmetric System

In the considered system, time is divided into equal slots and slots are numbered from 0. The length of a slot is assumed to be the minimum time required for any two nodes to discover each other and exchange the necessary information. Figure 4 describes my system model structure that represents $k$ slots difference asynchronous rendezvous scenario. The length of a node's CH sequence period is assumed to be $2M - 1$ as will be addressed in the following. Thus, $u = \{u_0, u_1, ..., u_i, ..., u_{2M-2}\}$ is a CH sequence, where $u_i \in [1, M]$ represents the channel index in the $i$th time slot of sequence $u$. Please note that the CH sequence changes every period. If I mark a slot number as $I$ and two CH sequences ($u$ and $v$) are given as shown in Figure 4, I say that node A and B rendezvous in the $i$th time slot on channel $c$ when $u_{\{i=I \bmod 2M-1\}} = v_{\{j=(I-k) \bmod 2M-1\}} = c$, where $c \in [1, M]$. In order words, any two nodes can rendezvous even though the misalignment distance of their starting slots is $k$, which is referred to as $k$-shift rendezvous [23]. In my proposed scheme, if an SU has data to transmit, it follows the transmit CH sequence; otherwise it follows the receive CH sequence, i.e., the sender and receiver have different CH sequences. In every round of the sending CH sequence, a random permutation of the available channels is used in the first $M$ time slots, and the reverse order of that random permutation excluding the last channel will be added in the next $M - 1$ time slots. Let the first part of $M$ time slots and the next part of $M - 1$ time slots denote *permutation part* and *reversed repetition part*, respectively. If I denote the permutation part with $R = \{r_0, r_1, ..., r_i, ..., r_{M-1}\}$, where $r_i \in [1, M]$, the sending CH sequence will be expressed as:

$$u_i = \begin{cases} r_i & \text{for} \quad 0 \leq i \leq M - 1 \\ r_{2M-2-i} & \text{for} \quad M \leq i \leq 2M - 2 \end{cases} \tag{1}$$

The receiving CH sequence selects one random channel among $M$ available channels and then stays on that channel for one round. Figure 5 illustrates an example of a rendezvous in FRARS when $M$ is 3. The transmit CH sequence represents three rounds and the receive CH sequence represents two rounds with $k$ slots difference from the sender, i.e., there is $k$ slots time difference between the sender and receiver. The shaded slots in the transmit CH sequence represents the permutation parts, and the rest of time slots represents the reversed repetition parts. For example, the sender visits channels 2 and 1 at slot numbers 3 and 4, which are in reverse order of slot numbers 0 and 1. Similarly the channel indexes in the slots numbered 5 and 6 are reused in slot numbers 8 and 9, while the receiving CH sequence randomly chose channel 1 in the first period and channel 3 in the second period. The shaded time slot of the received CH sequence refers to the rendezvous.

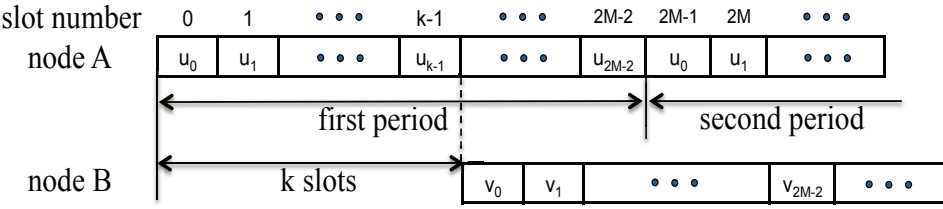

**Figure 4.** System model structure.

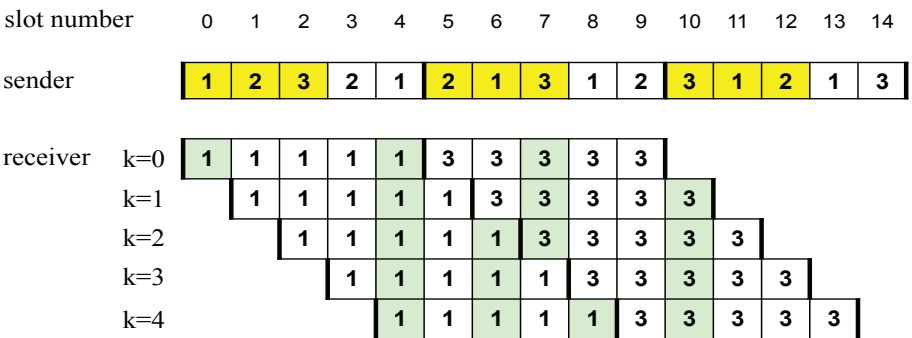

**Figure 5.** Illustration of FRARS when the number of available channels is 3.

**Theorem 1.** *Given M available channels, FRARS guarantees k-shift rendezvous for all $k(= 0, 1, ..., 2M - 2)$ so that any pair of sender and receiver must rendezvous within $2M - 1$ slots.*

**Proof.** For $k = 0$, the random permutation part $(u_0, u_1, ..., u_{M-1})$ in a sending CH sequence is totally overlapped with a receiving CH sequence, i.e., $TTR \leq M$. For $k = 1$, the first time slot of the permutation part is not overlapped with a receiving CH sequence, but the channel index $u_0$ is reused in the last time slot of the reversed repetition part $(u_0 = u_{2M-2})$. Therefore, the sender and receiver CH sequences $u$ and $v$ rendezvous in the worst case on channel $c$ when $u_{2M-2} = v_{2M-3} = c$, where $c \in [1, M]$, i.e., $TTR \leq 2M - 2$. For $k = 2$, in a similar way, the sender and receiver CH sequences $u$ and $v$ rendezvous in the worst case on channel $c$ when $u_{2M-2} = v_{2M-4} = c$, where $c \in [1, M]$. That is, $TTR \leq 2M - 3$. As the value of $k$ increases, the upper bound of TTR decreases until the value of $k$ is equal to $M - 1$, where $TTR \leq M$. For $k = M$, only $M - 1$ ending slots in the reversed repetition part are overlapped with a receiving CH sequence. Hence, rendezvous can occur in the next $M$ permutation part of the second period, i.e., $TTR \leq 2M - 1$. For $k = M + 1$, only $M - 2$ ending slots in the reversed repetition part are overlapped, thus the next $M$ permutation part of the second period is included in TTR, i.e., $TTR \leq 2M - 2$. The upper bound of TTR decreases until the maximum value of $k = 2M - 2$. Therefore, given $M$ available channels, the upper bound of TTR can be expressed as:

$$MTTR = \begin{cases} M & \text{for } k = 0 \\ 2M - 1 - (k \bmod M) & \text{for } 1 \leq k \leq 2M - 2 \end{cases} \quad (2)$$

Consequently, the upper bound of MTTR in FRARS is 2N-1. □

**Theorem 2.** *Given M available channels, ETTR of FRARS is not greater than $\frac{3}{4}M + \frac{1}{4M}$.*

**Proof.** For $k = 0$, since the random permutation part is totally overlapped with a receiving CH sequence, rendezvous can happen in each time slot within the permutation part with probability $1/M$, i.e., ETTR is $1\frac{1}{M} + 2\frac{1}{M} + 3\frac{1}{M} + ... + M\frac{1}{M} = \frac{M+1}{2}$. For $k = 1$, rendezvous can happen in the beginning $M - 1$ time slots of the receiving CH sequence $(v_0, v_1, ..., v_{M-2})$ with probability $1/M$ and one more rendezvous is possible in the last time slot of the reversed repetition part $(u_{2M-2} = v_{2M-3})$ with probability $1/M$, i.e., ETTR is $\frac{1}{M}\frac{(M-1)M}{2} + \frac{1}{M}(2M - 2)$. For $k = 2$, in a similar way, $M - 2$ time slots $(v_0, v_1, ..., v_{M-3})$ and two time slots $(v_{2M-5}, v_{2M-4})$ are possible rendezvous slots with probability $1/M$, i.e., ETTR is $\frac{1}{M}\frac{(M-2)(M-1)}{2} + \frac{1}{M}(2M - 3 + 2M - 4)$. Therefore, I can derive a generalized form of ETTR $0 \leq k \leq M - 1$ as follows:

$$ETTR = \frac{(M - k)(M - (k - 1))}{2M} + \frac{\sum_{i=1}^{k}(2M - (k + i))}{M}. \quad (3)$$

□

For $k = M$, rendezvous can happen in the overlapped $M - 1$ ending slots in the reversed repetition part with probability $1/M$, and one more rendezvous is possible in the next $M$ permutation slots of the second period with probability $1/M^2$, i.e., ETTR is $\frac{1}{M}\frac{(2M-(k+1))(2M-k)}{2} + \frac{\sum_{i=0}^{M-1}(2M-k+i)}{M^2}$. For $k = M+1$, in a similar way, rendezvous can happen in $M - 2$ ending slots with probability $1/M$ and two channels can rendezvous in the next $M$ permutation slots with probability $1/M^2$. Therefore, a generalized form of ETTR for $M \leq k \leq 2M - 2$ can be expressed as:

$$ETTR = \frac{(2M-(k+1))(2M-k)}{2M} + \frac{\sum_{i=0}^{M-1}(2M-k+i)}{M^2} \times (k-M+1). \tag{4}$$

To find an upper bound of ETTR, I rearrange Equations (3) and (4) as follows:

$$ETTR = \begin{cases} -\frac{k^2}{M} + (1 - \frac{1}{M})k + \frac{M+1}{2}, & 0 \leq k \leq M-1 \\ -\frac{k^2}{2M} + (\frac{3}{2} - \frac{1}{M})k - (\frac{M^2-4M+1}{2M}), & M \leq k \leq 2M-2 \end{cases} \tag{5}$$

Since the second derivative of (5) is negative, the value of $k$ for which the first derivative of (5) with respect to $k$ is equal to zero corresponds to a maximum value of ETTR, i.e., the maximum value of ETTR for $0 \leq k \leq M - 1$ is $\frac{3}{4}M + \frac{1}{4M}$ when $k = \frac{M-1}{2}$, and the maximum value of ETTR for $M \leq k \leq 2M - 2$ is $\frac{5}{8}M + \frac{1}{2}$ when $k = \frac{3M-2}{2}$. Since I assume that the value of $k$ is a positive integer value as depicted in Figure 4, the maximum value of ETTR for $0 \leq k \leq M - 1$ is equal to or greater than the maximum value of ETTR for $M \leq k \leq 2M - 2$. Consequently, the upper bound of ETTR in FRARS is $\frac{3}{4}M + \frac{1}{4M}$.

## 4.2. FRARS for Asymmetric System

I extend the FRARS algorithm to a general asymmetric algorithm by allowing each SU to have different available channels. When the geographical locations of the SUs are far, it is very likely that each SU has different available channels because of their relative locations to PUs. Suppose that the number of available channels for each sender and receiver are $|m_1|$ and $|m_2|$, the total number of channels is $M$, and the number of common available channels is $G$. The sender and receiver do not know each other's available channels, but at least one common channel is assumed to be available, otherwise rendezvous is impossible. Each SU must take total $M$ channels for each round of its CH sequence, thus the length of one round is $2M - 1$. In the first round of the receiver CH sequence, the randomly selected one channel among $m_2$ can rendezvous within $2M - 1$ time slots; however, it may not belong to $m_1$. Then the rendezvous cannot be achieved. However, the receiver does not have to consider unavailable channels that are not in $m_2$, and the rendezvous can be guaranteed within $|m_2|$ rounds of CH sequence, therefore the upper bound of MTTR for asymmetric system is $(2M - 1)(|m_2| + 1 - G)$. When the rendezvous happens in the first round of receiver's CH sequence, in other words, the receiver selects one of the commonly available channels in the first round of CH sequence, and the TTR value will be the same as the symmetric ETTR value $ETTR_{sym} = \frac{3}{4}M + \frac{1}{4M}$. For the second-round and third-round rendezvous cases, the TTR values are $(2M - 1) + ETTR_{sym}$ and $(2M - 1)2 + ETTR_{sym}$, respectively. Therefore, the expected TTR value for the asymmetric model can be computed as:

$$ETTR_{asym} = \sum_{i=0}^{|m_2|-G} \left( (2M-1)i + ETTR_{sym} \right) \cdot \frac{1}{|m_2| + 1 - G} \tag{6}$$

where $\frac{1}{|m_2|+1-G}$ is the probability of rendezvous of each round. This can be further simplified as $\frac{(2M-1)(|m_2|-G)}{2} + ETTR_{sym}$.

In the enhanced FRARS system, I also implement a replacement process to increase rendezvous probability by replacing unavailable channels in the CH sequence with randomly selected available

channels. In the EJS system, I investigate that the distribution of replacement channel selection is not uniformly distributed, and hence the performance of the asymmetric EJS system can be degraded. For example, suppose that $M = \{1, 2, 3, 4, 5\}$, $m_1 = \{2, 4\}$, and $G = \{4\}$. Then, according to the replacement algorithm, the unavailable channels 1, 3, 5 will be replaced by channel 2, thus the rendezvous probability is not improved. In addition, it is very likely that the performance of the EJS system can be degraded under a jamming attack.

To avoid this problem, I introduce a random-selection scheme for the replacement process in the enhanced FRARS system, i.e., the unavailable channels are replaced with randomly selected channels among the available channels. Using the same example as above, the unavailable channels 1, 3, 5 are replaced with randomly selected channels among $m_1 = \{2, 4\}$. Consequently, the probability of rendezvous will be enhanced by the replacement process. While the upper bound of MTTR for asymmetric system will be the same, the $ETTR_{asym}$ value is decreased by that random replacement scheme. Since the probability of rendezvous of each round, $\frac{1}{|m_2|+1-G}$, does not depend on $m_1$, the random replacement process can only affect $ETTR_{sym}$ value in the whole $ETTR_{asym}$ value in (6). To compute the modified $ETTR_{sym}$, I need to consider two different rendezvous cases: case 1—rendezvous on a randomly selected replaced channel; case 2—rendezvous on a non-replaced channel. Let $P_R$ denote the probability of case 1 and the ETTR values for each case are denoted as $ETTR_{case1}$ and $ETTR_{case2}$, respectively. The probability $P_R$ simply represents the number of random replaced channels over the whole one round time slots $2M - 1$. However, the number of random replaced channels can be either $2(M - |m_1|)$ or $2(M - |m_1|) - 1$ according to whether the last channel of the permutation part is included in the replaced channels. The probability of choosing $M - |m_1|$ channels including the last channel of the permutation part out of $M$ channels is $\frac{M-1}{{}_M C_{M-|m_1|}}$, hence the probability $P_R$ can be computed as:

$$P_R = \left(1 - \frac{M-1}{{}_M C_{M-|m_1|}}\right) \cdot \frac{2(M - |m_1|)}{2M - 1} + \frac{M-1}{{}_M C_{M-|m_1|}} \cdot \frac{2(M - |m_1|) - 1}{2M - 1}. \tag{7}$$

Since the $ETTR_{case2}$ is the same as $ETTR_{sym}$, I need to figure out $ETTR_{case1}$ to derive the final ETTR value. If the random variable $X$ is defined as a random independent sample extracted from $m_1$, then the probability of choosing one channel $\mathbb{P}\{X = x\}$ is $\frac{1}{|m_1|}$ for any $x \in m_1$. Thus, the probability of selecting the rendezvous channel is expressed as:

$$\mathbb{P}\{X = x\} = \frac{G}{|m_1|}, \forall x \in G. \tag{8}$$

Now, I denote the number of attempts at $R$ before the first rendezvous occurs. Then the rendezvous probability $\mathbb{P}\{R = n\}$ can be computed by:

$$\mathbb{P}\{R = n\} = (1 - p)^{n-1} p, \tag{9}$$

where $n$ is the $n$-th independent trial and $p$ is the success probability, which is equal to $\mathbb{P}\{X = x\} = \frac{G}{|m_1|}$. Therefore, the expected TTR value for case 1 can be expressed as:

$$ETTR_{case1} = \sum_{k=1}^{\infty} k(1-p)^{k-1} p = \frac{1}{p} = \frac{|m_1|}{G}. \tag{10}$$

Consequently, using (7) and (10) the modified $ETTR_{sym}$ can be determined by:

$$ETTR_{new\,sym} = P_R \cdot ETTR_{case1} + (1 - P_R) \cdot ETTR_{sym}. \tag{11}$$

If I take the lower bound of $P_R$ (i.e., $P_R = \frac{2(M-|m_1|)-1}{2M-1}$) in order to simplify the expression of the final ETTR, Equation (6) can be rewritten as $\frac{(2M-1)(|m_2|-G)}{2} + \left(\frac{2(M-|m_1|)-1}{2M-1} \cdot \frac{|m_1|}{G} + (1 - \frac{2(M-|m_1|)-1}{2M-1})\right) \cdot$

$\left(\frac{3}{4}M + \frac{1}{4M}\right)$. Numerical results show that the enhanced FRARS system outperforms the asymmetric EJS system. In particular, when the $G$ is small, the performance is more excellent. Moreover, the robustness of my proposed scheme against jamming attack is significantly higher than that of the EJS scheme.

## 5. Performance Evaluation

The considered network consists of two SUs and one channel-detection jammer that has similar functions as a legitimate user. Each SU is assumed to be equipped with a single antenna and operates in half-duplex mode, hence, no terminal can transmit and receive data simultaneously. However, the jammer can listen on two channels at the same time by using two transceivers to show how to quickly detect CH sequences of SUs. I implemented CH sequences for the sender and the receiver according to the considered rendezvous algorithms and the CDJA model by using MATLAB [10], and computed the TTR values and the rendezvous probabilities for the performance evaluation. I first evaluated the effect of the CDJA model on the EJS rendezvous method since the EJS system is an advanced system of the JS system. I then implemented the symmetric and asymmetric FRARS schemes to show that it is more resistant to jamming attacks than EJS systems.

First, in a symmetric scenario, the mean time to rendezvous (TTR) without considering a jamming attack is compared for both EJS and FRARS schemes. Figure 6 shows the mean TTR for both EJS and FRARS schemes when the $M$ increases from 4 to 100. For each available channel case, I repeated the above process 1000 times to calculate the average TTR. The starting position of the receiver can be any time slot of the sender's sequence in the both EJS and FRARS schemes due to no time synchronization. In both schemes, as the number of available channels increases, the average TTR tends to gradually increase. The TTR result of symmetric EJS is about half value of $M$ and is slightly better than that of the symmetric FRARS. The results are consistent with the upper-bound ETTR of the symmetric EJS scheme (i.e., $\frac{3}{2}P + 3$) [9] and the ETTR of the FRARS scheme (i.e., $\frac{3}{4}M + \frac{1}{4M}$) respectively.

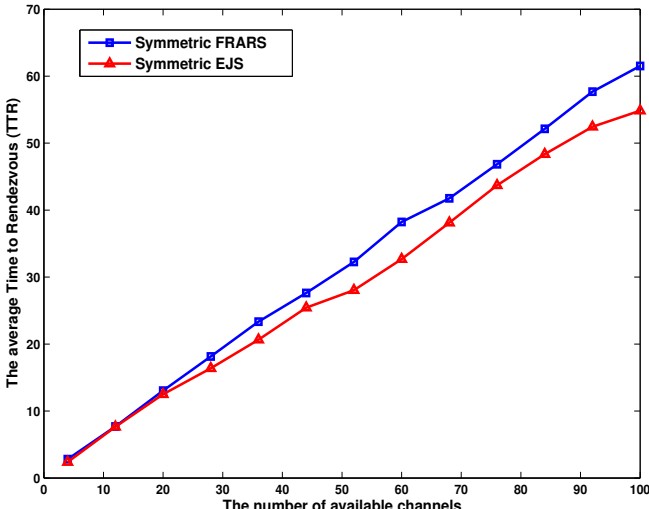

**Figure 6.** The average Time To Rendezvous (TTR) results for the symmetric FRARS and EJS schemes with $M = [4, \cdots, 100]$.

Next, I compare the rendezvous probabilities for symmetric EJS and FRARS schemes under jamming attacks using the CDJA model. For the EJS scheme, a single jammer can fully detect the CH sequence of a sender within the $P$ time slots and jams all remaining time slots; thus, I use two jammers for the FRARS scheme in this experiment. As shown in Figure 7, the rendezvous probability of

symmetric EJS scheme is significantly reduced due to the CDJA attack and is less than 10%. However, the probability of a rendezvous in a symmetric FRARS system is almost stable and close to 100%.

For an asymmetric scenario, in Figure 8, I show the average TTR of the asymmetric FRARS as well as the theoretical ETTR of FRARS mentioned in Section 4.2. The number of available channels of the sender or receiver is denoted by $m_i$ and is half of the $M$ (i.e., $|m_i|/|M| = 0.5$) and $|G| = 1$ for various $M = \{10, 20, \cdots, 90, 100\}$. Figure 8 shows that the average TTR values for the asymmetric FRARS are not significantly different from the theoretical ETTR $\frac{(2M-1)(|m_2|-G)}{2} + \left( \frac{2(M-|m_1|)-1}{2M-1} \cdot \frac{|m_1|}{G} + \left(1 - \frac{2(M-|m_1|)-1}{2M-1}\right) \cdot \left(\frac{3}{4}M + \frac{1}{4M}\right)\right)$.

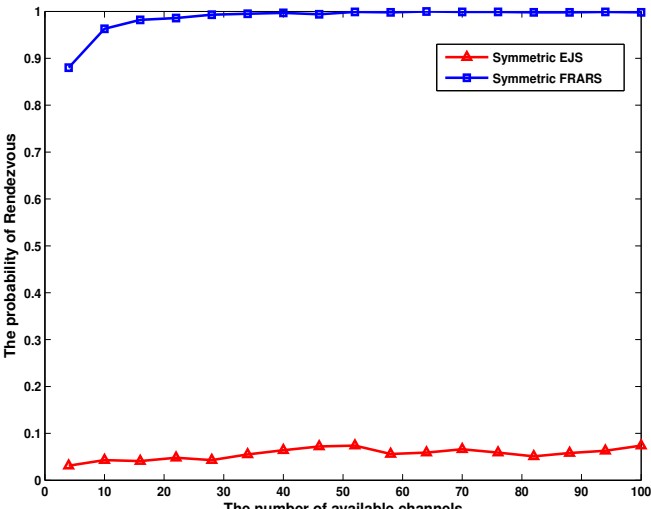

**Figure 7.** The probability of rendezvous for the symmetric EJS and FRARS schemes under CDJA attacks with $M = [4, \cdots, 100]$.

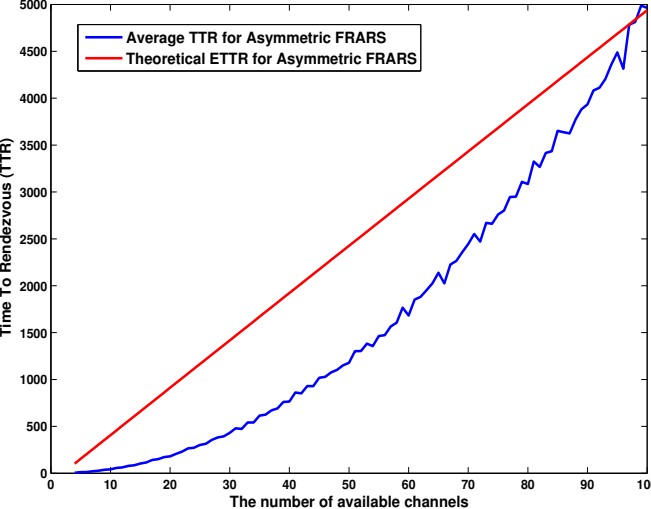

**Figure 8.** The average TTR and the theoretical ETTR for the asymmetric FRARS scheme with $|m_1| = |m_2| = |M|/2$, $M = [4, \cdots, 100]$, and $G = 1$.

As I have seen in the above symmetric scenarios, I now firstly analyze the asymmetric EJS and FRARS schemes by comparing the mean TTRs without a jamming attack. In addition, I then compare the rendezvous probability for both the asymmetric EJS and FRARS using the CDJA model. Figure 9 shows the average TTRs for the asymmetric EJS and FRARS schemes when the $M$ is

$\{10, 20, \cdots, 90, 100\}$, $|m_1| = |m_2| = |M|/2$, and $G$ are 1 and 10. When the $G$ is 1, the average TTR of the asymmetric FRARS is about half of the asymmetric EJS algorithm. However, as the $G$ increases, the average TTRs of both asymmetric FRARS and EJS schemes are similar to each other. Moreover, it is clear that the average TTRs for both EJS and FRARS schemes significantly decrease when the $G$ increases.

Finally, for the jamming attack in an asymmetric scenario, Figure 10 shows the rendezvous probability of the asymmetric EJS and FRARS schemes under the CDJA attacks when the ratio $|m_i|/|M| = 0.5$ and $M = \{40, 100\}$. Under the CDJA attack, it is clear that the asymmetric EJS system's rendezvous probability decreases dramatically regardless of the value of $M$. As the number of common channels increases to $G = 20$, the probability of rendezvous increases slightly, and it does not exceed 15%. However, the rendezvous probability of the FRARS schemes for both the $M = 40$ and $M = 100$ cases are 100% within the maximum time slots that was the duration of the runs for this experiment. The reason for this is that in FRARS scheme, since the repetition of the CH sequence of the sender is independent of the previous repetition, it is almost impossible for the jammer to estimate the CH sequence of the sender.

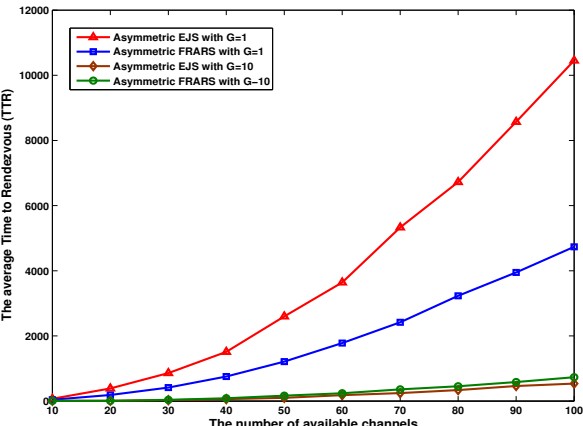

**Figure 9.** The average Time to Rendezvous (TTR) for the asymmetric FRARS and EJS schemes with $|m_1| = |m_2| = |M|/2$, $M = \{10, 20, \cdots, 90, 100\}$, and $G = \{1, 10\}$.

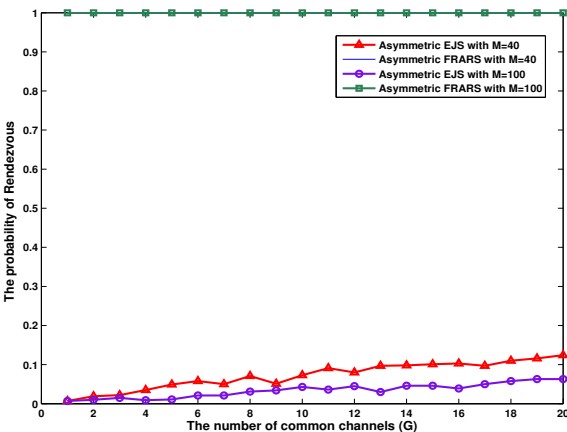

**Figure 10.** The probability of rendezvous for the asymmetric FRARS and EJS schemes under CDJA attacks with $|m_1| = |m_2| = |M|/2$, $M = \{40, 100\}$, and $G = \{1 \cdots 20\}$.

## 6. Conclusions

In this paper, I examined the vulnerability of the well-known asymmetric rendezvous schemes such as JS and EJS under a sophisticated jamming attack. By using the CDJA model, the sender's CH

sequences in JS and EJS are fully detected and jammed completely after a certain time slot. To enhance robustness under a sophisticated jamming attack, I proposed a FRARS algorithm that can apply the randomized permutation technique in the CH sequences. Moreover, I showed that my proposed FRARS algorithm can be easily extended to a general asymmetric scenario with replacement scheme. The upper bounds of MTTR and ETTR of the FRARS for both symmetric and asymmetric systems are derived, and my numerical results showed that the rendezvous probabilities for the symmetric and asymmetric FRARS systems are close to 100% while the rendezvous probabilities for the EJS systems are less than 15%. Therefore, my proposed FRARS algorithm vastly outperforms other asymmetric rendezvous schemes under a sophisticated jamming attack.

**Funding:** This research received no external funding.

**Acknowledgments:** This research was supported by Hwarang-dae research institute of Korea Military Academy.

**Conflicts of Interest:** The authors declare no conflict of interest.

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
