# Peer review of "Fast and Robust Asynchronous Rendezvous Scheme for Cognitive Radio Networks"

_applsci, doi:10.3390/app9122481_

Round 1

Reviewer 1 Report

The author examined the vulnerability of EJS under a sophisticated jamming attack. To enhance robustness, he proposed a FRARS algorithm. Simulation results showed that his proposal outperforms  EJS. The author should clearly explain why he has chosen EJS (EJS was proposed in 2013). The related work should be improved (references are old). the author would add a glossary because there are many abbreviations. 

Author Response

Response to Reviewer 1 Comments

Point 1: The author examined the vulnerability of EJS under a sophisticated jamming attack. To enhance robustness, he proposed a FRARS algorithm. Simulation results showed that his proposal outperforms EJS. The author should clearly explain why he has chosen EJS (EJS was proposed in 2013).

Response 1: Thank you for your comment. The reason that I have chosen EJS rendezvous algorithm as a conventional one to compare with my proposed algorithm is that the EJS is known as one of the best blind rendezvous algorithms for CRNs based on non-deterministic CH sequences and the guaranteed rendezvous times for both symmetric and asymmetric models. I added this explanation in the Introduction section.

Point 2: The related work should be improved (references are old). The author would add a glossary because there are many abbreviations.

Response 2: Thank you for your comment. I added three more recently published papers in the reference section and modified the related work section accordingly. I believe that your advice significantly improved the quality of this article.

Reviewer 2 Report

In the introduction, due to how the initial part is structured, the general feeling for the reader is that Cognitive Radios and the use of licensed spectrum on a shared basis is an exclusive of the FCC. Since this is not the case, I would suggest to re-formulate this section to clearly reflect that this is a worldwide aspect. The author states that the control channel is likely to be occupied by a primary user. I would re-formulate this sentence as "probable" instead of "likely," since there is no certainty on this. The literature review is sufficient for rendezvous approaches, but definitely lacking for CRs. This has been an extensively addressed research field in the last 10 years, so the author should improve the introduction taking this into account. Although the mathematical formulation is clear and the numerical results show a good performance of the proposed solution, there is a significant lack of information on the system setup. What is the frequency range for the simulations? Is there any channel model that might impact the performance? These aspects shall be carefully addressed; even if there is no impact, this should be clarified in the paper.

Author Response

Response to Reviewer 2 Comments

Point 1: In the introduction, due to how the initial part is structured, the general feeling for the reader is that Cognitive Radios and the use of licensed spectrum on a shared basis is an exclusive of the FCC. Since this is not the case, I would suggest to re-formulate this section to clearly reflect that this is a worldwide aspect.

Response 1: Thank you very much for your thoroughness, I strongly agree that Cognitive Radios and the use of licensed spectrum on a shared basis is a worldwide aspect. To clearly reflect this point, I modified the beginning sentences in the introduction section.

Point 2: The author states that the control channel is likely to be occupied by a primary user. I would re-formulate this sentence as "probable" instead of "likely," since there is no certainty on this.

Response 2: Thank you for your comment. I changed the term "likely"  to "probably" since there is no certainty on that.

Point 3: The literature review is sufficient for rendezvous approaches, but definitely lacking for CRs. This has been an extensively addressed research field in the last 10 years, so the author should improve the introduction taking this into account.

Response 3: I agree that CRs have been extensively studied over the last decade. Therefore, I added two more CR related papers in the reference section and I added more explanations about CRs in the introduction section.

Point 4: Although the mathematical formulation is clear and the numerical results show a good performance of the proposed solution, there is a significant lack of information on the system setup.

Response 4: I apologize for not clearly explaining the system setup. The performance evaluation section is based on numerical analysis and not on real simulations. First, I created CH sequences for the sender and the receiver according to the considered rendezvous algorithms, then computed the TTR value under no jamming attack and the rendezvous probability under CDJA. Second, I repeated the above process 1000 times with different CH sequences in order to calculate the average values. I changed the term "simulation results"  to "numerical results" to avoid confusion in this paper and added more detailed explanations in the performance evaluation section.

Point 5: What is the frequency range for the simulations? Is there any channel model that might impact the performance? These aspects shall be carefully addressed; even if there is no impact, this should be clarified in the paper.

Response 5: Thank you for your comment. I apologize for not clearly addressing the frequency range and channel model issue in the considered network. As mentioned above, the performance evaluation section is based on numerical analysis and not on real simulations. Thus I did not consider the real frequency range for the channels, and the impact of channel model was not considered in this paper. I agree that channel models can have an impact on performance in real simulation environments. In my future work, I will definitely consider those issues. I believe that your comments significantly improved the quality of this article.   

Reviewer 3 Report

The paper is globally almost well written and organized. However, as the large part of research work in Computer Science it presents the usual weakness: it is full of mathematical concepts and proofs but only a very little percentage of the work is dedicated to experimental section and simulations results; more specifically, too much equations, too few details about simulations have been provided, no reference standards, no simulation parameter set table, no examples about the used tool (authors only mentioned that is a sort of Matlab external tool). Author should take into account this aspect while balancing experimental and theoretical sections. Therefore, in my opinion, also English level used in the paper should be improved.

Author Response

Response to Reviewer 3 Comments

Point 1: The paper is globally almost well written and organized. However, as the large part of research work in Computer Science it presents the usual weakness: it is full of mathematical concepts and proofs but only a very little percentage of the work is dedicated to experimental section and simulations results; more specifically, too much equations, too few details about simulations have been provided, no reference standards, no simulation parameter set table, no examples about the used tool (authors only mentioned that is a sort of Matlab external tool). Author should take into account this aspect while balancing experimental and theoretical sections.

Response 1: I apologize for not clearly explaining the performance evaluation. The performance evaluation section is based on numerical analysis and not on real simulations. First, I created CH sequences for the sender and the receiver according to the considered rendezvous algorithms, then computed the TTR value under no jamming attack and the rendezvous probability under CDJA. Second, I repeated the above process 1000 times with different CH sequences in order to calculate the average values. I changed the term "simulation results"  to "numerical results" to avoid confusion in this paper and added more detailed explanations in the performance evaluation section.

Point 2: Therefore, in my opinion, also English level used in the paper should be improved.

Response 2: Thank you for your comment. I took your advice and have revised the entire manuscript with the support of a proofreading expert.

Round 2

Reviewer 2 Report

The author addressed the comments provided in the previous revision. I have no other requests for clarification.